# Memory CD8 T cells are vulnerable to chronic IFN-γ signals but not to CD4 T cell deficiency in MHCII-deficient mice

Ruka Setoguchi ®[1,2] ✉, Tomoya Sengiku[2], Hiroki Kono[2], Eiryo Kawakami ®[3,4,5,6], Masato Kubo ®[7,8], Tadashi Yamamoto ®[1,9] & Shohei Hori ®[2,10]

The mechanisms by which the number of memory CD8 T cells is stably maintained remains incompletely understood. It has been postulated that maintaining them requires help from CD4 T cells, because adoptively transferred memory CD8 T cells persist poorly in MHC class II (MHCII)-deficient mice. Here we show that chronic interferon-γ signals, not CD4 T cell-deficiency, are responsible for their attrition in MHCII-deficient environments. Excess IFN-γ is produced primarily by endogenous colonic CD8 T cells in MHCII-deficient mice. IFN-γ neutralization restores the number of memory CD8 T cells in MHCII-deficient mice, whereas repeated IFN-γ administration or transduction of a gain-of-function STAT1 mutant reduces their number in wild-type mice. CD127[high] memory cells proliferate actively in response to IFN-γ signals, but are more susceptible to attrition than CD127[low] terminally differentiated effector memory cells. Furthermore, single-cell RNA-sequencing of memory CD8 T cells reveals proliferating cells that resemble short-lived, terminal effector cells and documents global downregulation of gene signatures of long-lived memory cells in MHCII-deficient environments. We propose that chronic IFN-γ signals deplete memory CD8 T cells by compromising their long-term survival and by diverting self-renewing CD127[high] cells toward terminal differentiation.

Memory CD8 T cells are essential for host protection from intracellular pathogens and malignant cells. Upon priming, antigen-specific, naïve CD8 T cells undergo extensive clonal expansion and differentiate into either KLRG1[high]CD127[low] short-lived, terminally differentiated effector (TE) cells or CD127[high] multipotent memory precursor effector (MP) cells[1–3]. After elimination of the antigen, most TE cells undergo apoptosis while MP cells give rise to memory CD8 T cells, which are then maintained for long periods[1–3]. Long-term maintenance of memory CD8 T cells is an important hallmark of immunological memory and a prerequisite for successful vaccination and cancer immunotherapy.

[1]Formerly Laboratory for Immunogenetics, RIKEN Center for Integrative Medical Sciences, Yokohama City, Kanagawa 230-0045, Japan. [2]Laboratory of Immunology and Microbiology, Graduate School of Pharmaceutical Sciences, The University of Tokyo, Tokyo 113-0033, Japan. [3]Advanced Data Science Project (ADSP), RIKEN Information R&D and Strategy Headquarters, RIKEN, Yokohama City, Kanagawa 230-0045, Japan. [4]Department of Artificial Intelligence Medicine, Graduate School of Medicine, Chiba University, Chiba 260-8670, Japan. [5]Institute for Advanced Academic Research (IAAR), Chiba University, Chiba 260-8670, Japan. [6]Chiba University Synergy Institute for Futuristic Mucosal Vaccine Research and Development (cSIMVa), Chiba University, Chiba 260-8670, Japan. [7]Division of Molecular Pathology, Research Institute for Biomedical Science, Tokyo University of Science, 2669 Yamazaki, Noda-shi, Chiba 278-0022, Japan. [8]Laboratory for Cytokine Regulation, RIKEN Center for Integrative Medical Sciences, Yokohama City, Kanagawa 230-0045, Japan. [9]Cell Signal Unit, Okinawa Institute of Science and Technology Graduate University, Okinawa 904-0495, Japan. [10]Formerly Laboratory for Immune Homeostasis, RIKEN Center for Integrative Medical Sciences, Yokohama City, Kanagawa 230-0045, Japan. ✉e-mail: ruka.setoguchi@mol.f.u-tokyo.ac.jp

The common γ-chain cytokines, IL-7 and IL-15, are required for survival and homeostatic proliferation of memory CD8 T cells, respectively[4–7]; however, mechanisms that control their long-term maintenance remain poorly understood.

Memory CD8 T cells are phenotypically and functionally heterogeneous[8]. Apart from tissue-resident memory cells[9], circulating memory T cells have traditionally been classified into central memory ($T_{CM}$) cells and effector memory ($T_{EM}$) cells, based on differential expression of the lymph node homing receptors, CD62L and CCR7[10,11]. CD62L$^{high}$CCR7$^{high}$ $T_{CM}$ cells and CD62L$^{low}$CCR7$^{low}$ $T_{EM}$ cells display distinct migratory, homeostatic, and functional properties. $T_{CM}$ cells are thought to possess higher proliferative potential, multipotency, and lymph node homing capacity, whereas $T_{EM}$ cells are believed to be more cytotoxic, and they continuously circulate through blood and non-lymphoid tissues[8]. However, recent studies have discovered heterogeneity of the CD62L$^{low}$ subpopulation and have shown that it includes a persistent, effector-like population of KLRG1$^{+12,13}$, CD127$^{low14}$ or CX3CR1$^{high}$ cells[15]. These effector-like memory cells originate mainly from TE cells rather than MP cells and are phenotypically and functionally distinct from $T_{EM}$ cells (KLRG1$^-$, CD127$^{high}$ or CXCR1$^{-/int}$ CD62L$^{low}$), as well as $T_{CM}$ cells. Unlike $T_{EM}$ and $T_{CM}$ cells, they display potent cytotoxicity, but have limited proliferation potential, durability, and multipotency[13–15]. Since these effector-like memory CD8 T cells display unique features shared by both effector cells and long-lived memory cells, they are distinguished from $T_{EM}$ cells and are designated as long-lived effector cells (LLECs)[13] or terminally differentiated $T_{EM}$ (t-$T_{EM}$) cells[14]. However, environmental cues that positively or negatively regulate homeostasis of these distinct memory CD8 T cell subsets remain ill-defined.

Using several CD4 T-cell-deficient mouse models, including mice treated with anti-CD4 depleting mAb, CD4-deficient mice, and MHCII-deficient mice[16–19], previous studies have established that CD4 T cells are required for secondary responses of CD8 T cells[20,21]. While CD4 T cell help during the priming phase is required for CD8 T cells to differentiate into functional memory cells, various studies have reached contradictory conclusions as to whether CD4 T cell help is also required for persistence of memory CD8 T cells. Sun et al. have shown that adoptively transferred memory CD8 T cells persist poorly in MHCII-deficient host mice[19,22]. In contrast, Shedlock et al. found that antigen-specific memory CD8 T cells can persist in CD4-deficient mice and anti-CD4 mAb-treated mice[18]. Later studies have also yielded conflicting results as to whether adoptively transferred memory CD8 T cells can persist in CD4-deficient host mice as stably as in wild-type (WT) host mice[23,24]. While the root of these apparently conflicting observations remains unclear, these studies agree on the severe attrition of memory CD8 T cells in MHCII-deficient environments[23,24]. Therefore, we hypothesized that an unrecognized factor, other than CD4 T cell deficiency, is responsible for defective persistence of memory CD8 T cells in MHCII-deficient mice. This study was undertaken to identify such a factor and to determine how MHCII-deficient environments impair homeostasis of heterogeneous memory CD8 T cell subsets. We demonstrate that elevated IFN-γ production primarily from endogenous colonic CD8 T cells, not CD4 T cell-deficiency, is responsible for attrition of memory CD8 T cells in MHCII-deficient mice. Furthermore, we provide evidence that MHCII-deficient environments selectively compromise maintenance of CD127$^{high}$ $T_{CM}$ and/or $T_{EM}$ cells via chronic IFN-γ signals by driving them into terminal differentiation rather than self-renewal upon proliferation.

## Results

### Defective maintenance of CD127$^{high}$ memory CD8 T cells in MHCII-deficient, but not in CD4 T cell-depleted mice

We first revisited the aforementioned transfer experiment and compared the number of adoptively transferred memory CD8 T cells in WT, MHCII$^{-/-}$ and CD4 T cell-depleted WT host mice. We transferred Thy1.1 naïve P14 TCR transgenic CD8 T cells, which specifically react with the LCMV GP$_{33–41}$ peptide presented by H-2D$^b$, into Thy1.2 WT host mice and infected them with vaccinia virus expressing this epitope (Vac-GP33). More than 30 days later, total CD8 T cells, including Thy1.1 P14 T cells, were transferred into either Thy1.2 MHCII$^{-/-}$ mice, WT mice treated with anti-CD4 depleting mAb (GK1.5) or non-treated WT mice (Fig. 1a). CD4 T cells were efficiently depleted upon GK1.5 treatments when assessed on days 3, 18 (Supplementary Fig. 1a) and 38 after the secondary transfer (Fig. 1b). The abundance of donor P14 T cells in peripheral blood lymphocytes (PBLs) was comparable among the three groups on days 3 and 18, but was reduced on day 38 in MHCII$^{-/-}$ mice, but not in GK1.5-treated WT mice, compared to non-treated WT mice (Fig. 1b, Supplementary Fig. 1a). Numbers of donor P14 T cells in the spleen and liver were also reduced in MHCII$^{-/-}$ mice, but not in GK1.5-treated WT mice on days 40–50 after the secondary transfer (Fig. 1b, Supplementary Fig. 1b). Furthermore, when CD4$^{-/-}$ mice were used as secondary recipients, we found no reduction in the number of P14 T cells on day 42 (Supplementary Fig. 1c). Thus, in agreement with a previous report[23], a significant reduction in the number of memory CD8 T cells was observed only in MHCII$^{-/-}$ mice.

We then compared the phenotype of P14 T cells and examined expression of the memory T cell-associated molecule, CD127, along with a TE cell- and t-$T_{EM}$ cell/LLEC-associated molecule, KLRG1. Consistent with previous reports[13,14], memory P14 T cells were heterogeneous for CD127 and KLRG1 and consisted of both CD127$^{high}$ and CD127$^{low}$ subsets, the majority of which were KLRG1$^-$ and KLRG1$^+$, respectively (Fig. 1c). The frequency of KLRG1$^+$CD127$^{low}$ cells was increased, while that of KLRG1$^-$CD127$^{high}$ cells was reduced, among P14 T cells from MHCII$^{-/-}$ mice (Fig. 1c, Supplementary Fig. 1d). As a result, the number of KLRG1$^-$CD127$^{high}$ cells was selectively reduced, whereas that of KLRG1$^+$CD127$^{low}$ cells remained unchanged, in MHCII$^{-/-}$ mice (Fig. 1c). The number of KLRG1$^+$CD127$^{high}$ and KLRG1$^-$CD127$^{low}$ cells was also reduced in MHCII$^{-/-}$ mice (Supplementary Fig. 1d). When assessed later (day 97 or 113), however, the number of KLRG1$^+$CD127$^{low}$ cells was also reduced in MHCII$^{-/-}$ mice, although extent of the reduction was less marked for KLRG1$^+$CD127$^{low}$ cells than for KLRG1$^-$CD127$^{high}$ cells (Supplementary Fig. 1e). Thus, CD127$^{high}$ memory cells are more susceptible to attrition than CD127$^{low}$ t-$T_{EM}$ cells/LLECs in MHCII-deficient environments.

We also noted reduced expression of CD127 on both KLRG1$^+$ and KLRG1$^-$ P14 T cells from MHCII$^{-/-}$ mice (Fig. 1d). Expression of another memory-associated molecule[25], CXCR3, was also downregulated in both subsets, whereas expression of CD27 was not (Fig. 1d), indicating that not only the number of CD127$^{high}$ cells, but also their CD127 and CXCR3 expression is selectively reduced in MHCII$^{-/-}$ mice. In addition, we noted downregulation of CD127 and CXCR3 on KLRG1$^-$, but not on KLRG1$^+$ P14 cells from GK1.5-treated WT mice (Fig. 1c, d), indicating that CD4 T cell-deficiency contributes to downregulation of CD127 and CXCR3 on CD127$^{high}$ memory cells.

These results collectively indicate that MHCII-deficiency selectively impairs persistence of CD127$^{high}$ memory CD8 T cells independently of CD4 T cell-deficiency, and suggests that another factor present only in MHCII$^{-/-}$ mice is responsible for their attrition.

### Elevated IFN-γ production is responsible for attrition of CD127$^{high}$ memory CD8 T cells, although it promotes their proliferation, in MHCII-deficient environments

To identify such a factor, we first examined expression levels of IL-7 and IL-15, which are required for memory T cell maintenance[4–7]. No difference was observed in *Il7* and *Il15* mRNA levels in spleens of MHCII$^{-/-}$ and WT mice (Supplementary Fig. 2a). Protein levels of IL-7 and the IL-15/IL-15Rα complex in the spleen were also comparable (Supplementary Fig. 2b).

To uncover molecular pathways perturbed in memory CD8 T cells by MHCII-deficient environments, we performed RNA sequencing

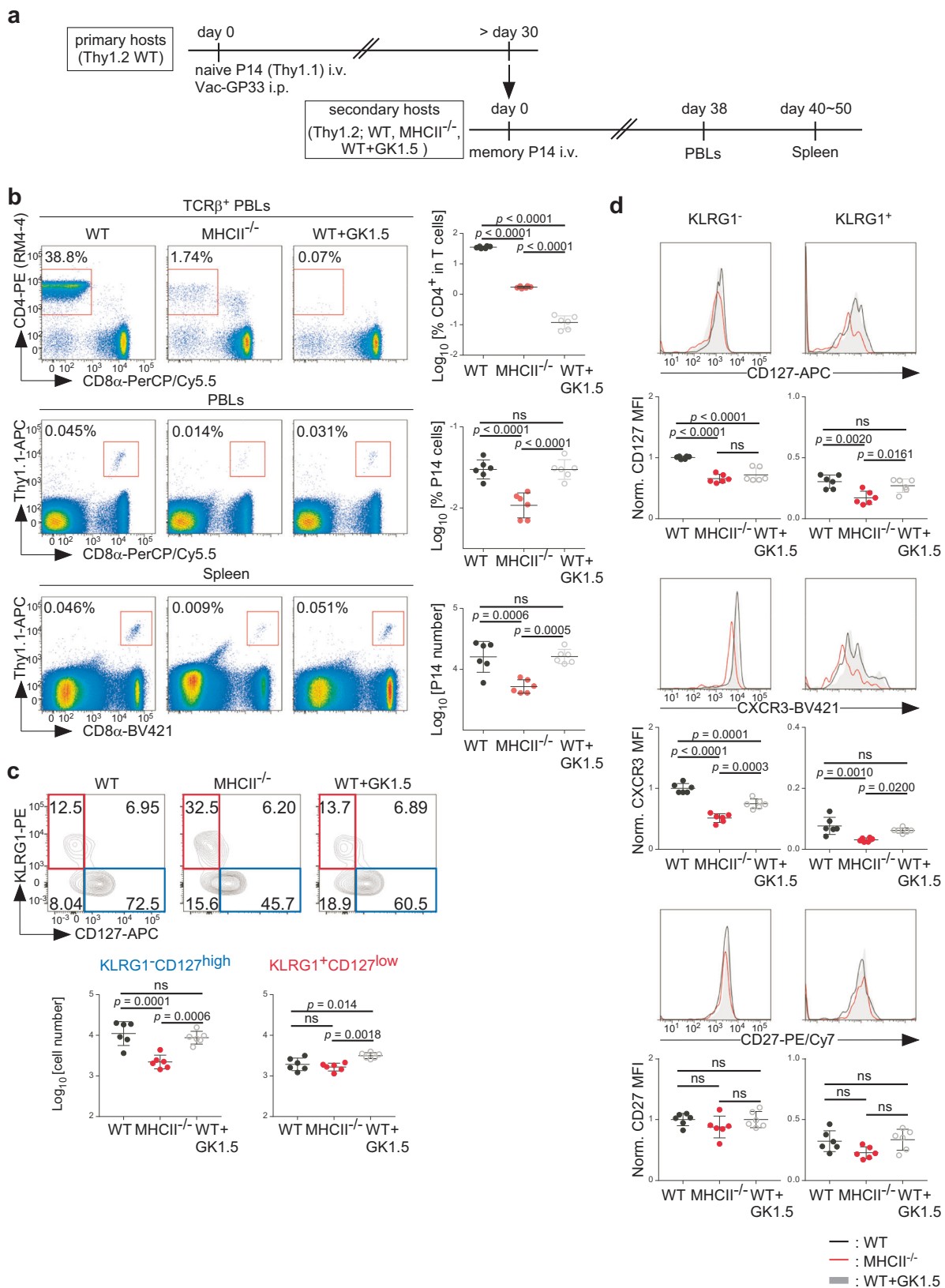

(RNA-seq) on memory P14 T cells isolated from MHCII⁻/⁻ host mice and those from WT host mice 2 weeks after the secondary transfer. We chose this early time point at which the number of donor P14 T cells had not yet significantly decreased, but CD127 expression on KLRG1⁻ P14 T cells was already downregulated in MHCII⁻/⁻ hosts (Supplementary Fig. 1a, 2c), to detect transcriptional changes that precede, and

possibly cause, impaired memory CD8 T cell maintenance in MHCII-deficient environments.

Differential expression analyses revealed 56 upregulated and 9 down-regulated genes in P14 T cells from MHCII⁻/⁻ mice (Fig. 2a, Supplementary Table 1). Down-regulated genes included *Il7r* (encoding CD127), consistent with the flow cytometry data (Fig. 2a,

**Fig. 1 | Defective maintenance of CD127^high memory CD8 T cells in MHCII-deficient, but not in CD4 T cell-depleted mice. a** Experimental scheme. Thy1.2 WT mice received Thy1.1 naïve P14 TCR transgenic CD8 T cells and were infected with Vac-GP33. More than 30 days later, total CD8 T cells containing Thy1.1 memory P14 T cells were transferred into Thy1.2 MHCII^−/− mice, WT mice treated with anti-CD4 mAb (GK1.5) or non-treated WT mice. PBLs and splenocytes were subjected to flow cytometric analysis on days 38 and 40–50 after the secondary transfer, respectively. **b** Representative flow cytometric profiles of TCRβ^+ PBLs (upper panels), total PBLs (middle panels) and total splenocytes (lower panels). The percentage of CD4^+ cells in TCRβ^+ PBLs, the percentage of Thy1.1^+CD8α^+Vα2^+ P14 T cells in total PBLs

and the number of Thy1.1^+CD8α^+Vα2^+ P14 T cells in the spleen are summarized (mean ± SD). **c** Representative flow cytometric profiles of Thy1.1^+CD8α^+Vα2^+ P14 T cells from the spleen. The number of indicated subsets is summarized (mean ± SD). **d** Representative histograms of CD127, CXCR3 and CD27 expression on KLRG1^− and KLRG1^+ Thy1.1^+CD8α^+Vα2^+ P14 T cells from the spleen. The geometric mean of fluorescence intensity (MFI) of each marker was normalized by the average value of KLRG1^− cells from WT mice and summarized (mean ± SD). Each symbol represents one mouse (*n* = 6 per group). Results are pooled from two independent experiments. Data were analyzed using one-way ANOVA with Tukey's multiple comparisons test. Source data are provided as a Source Data file.

Supplementary Fig. 2c). When these differentially expressed genes (DEGs) were subjected to Gene Ontology (GO) enrichment analysis, genes upregulated in P14 T cells from MHCII^−/− mice were enriched for "immune response-related pathways" and "cell cycle-related pathways" (Fig. 2b). Notably, the former included GO terms "response to interferon (IFN)-α", "response to IFN-β" and "response to IFN-γ" (Fig. 2b). In addition, *Stat1*, which encodes the major transcription factor activated and upregulated in response to type I or II IFNs[26,27], was upregulated in P14 T cells from MHCII^−/− host mice (Fig. 2a). These results suggest that type I and/or II IFN signals are elevated in memory CD8 T cells from MHCII^−/− mice.

Consistent with up-regulation of IFN-induced genes, plasma concentrations of IFN-γ, but not IFN-β, were significantly elevated in MHCII^−/−, but not in CD4^−/− mice compared to WT mice (Fig. 2c, Supplementary Fig. 2b). By contrast, IFN-γ concentrations were not affected in CD4^−/− mice (Fig. 2c). Increased IFN-γ production is therefore a phenotype of MHCII^−/− mice that is independent of CD4 T cell-deficiency. To evaluate whether elevated IFN-γ levels are responsible for impaired maintenance of memory CD8 T cells in MHCII^−/− mice, we transferred memory P14 T cells into MHCII^−/− and WT host mice treated with or without anti-IFN-γ neutralizing mAb. The number of P14 T cells was comparable between anti-IFN-γ-treated and untreated MHCII^−/− mice on day 14 after transfer (Supplementary Fig. 2d). However, when assessed on days 40–45, the number of P14 T cells was fully restored in MHCII^−/− mice by anti-IFN-γ treatments, whereas it was not affected in WT mice (Fig. 2d).

The number of KLRG1^−CD127^high and KLRG1^+CD127^low P14 T cells was also increased by IFN-γ neutralization in MHCII^−/− mice, although the extent of the increase was more marked for KLRG1^−CD127^high cells than for KLRG1^+CD127^low cells (Fig. 2e, Supplementary Fig. 2e). Expression of CD127 and CXCR3, but not CD27, on both KLRG1^+ and KLRG1^− P14 T cells was also upregulated by anti-IFN-γ treatments in MHCII^−/− mice (Fig. 2f). Thus, in addition to CD4 T cell-deficiency, elevated IFN-γ concentrations also contribute to downregulation of CD127 and CXCR3 on memory P14 T cells in MHCII^−/− mice. Collectively these data indicate that increased IFN-γ production is primarily responsible for attrition of CD127^high memory CD8 T cells in MHCII-deficient environments.

Our RNA-seq analysis also revealed upregulation of genes of cell cycle-related pathways, including *Mki67* (encoding Ki67), in memory P14 T cells from MHCII^−/− mice (Fig. 2a, b). Indeed, the abundance of Ki67^+ cells among memory P14 T cells, especially in KLRG1^−CD127^high P14 T cells, was higher in MHCII^−/− mice than in WT mice (Fig. 2g). This increased abundance of Ki67^+ cells in MHCII^−/− mice was reduced by IFN-γ neutralization (Fig. 2h). Thus, elevated IFN-γ levels impair memory CD8 T cell maintenance despite promoting their proliferation in MHC^−/− mice.

**Repeated IFN-γ administration impairs maintenance of memory CD8 T cells while promoting their proliferation in the presence of CD4 T cells**

We then asked whether increased IFN-γ levels impair maintenance of memory CD8 T cells in the presence of CD4 T cells. To this end, Thy1.2 WT mice that had received Thy1.1 naïve P14 T cells and had been

primed with Vac-GP33 were repeatedly administered recombinant murine IFN-γ (rmIFN-γ) >30 days post-infection. Frequencies of P14 T cells in PBLs declined progressively and more markedly in rmIFN-γ-treated mice than in PBS-treated mice (Fig. 3a). On days 24–25 after the first rmIFN-γ injection, the number of memory P14 T cells in the spleen was reduced in rmIFN-γ-treated mice (Fig. 3b). Moreover, the percentage of Ki67^+ cells among P14 T cells increased significantly upon rmIFN-γ treatments (Fig. 3c). In contrast, a single injection of rmIFN-γ into WT host mice failed to affect the percentage of memory P14 T cells in blood or their number in the spleen (Supplementary Fig. 3a).

While the number of KLRG1^+CD127^low and KLRG1^−CD127^high memory P14 T cell subsets was reduced by rmIFN-γ treatments, the extent of the reduction was less marked for KLRG1^+CD127^low cells than for KLRG1^−CD127^high cells (Fig. 3d, Supplementary Fig. 3b). Expression of CD127, but not CD27 was downregulated on both KLRG1^+ and KLRG1^− memory P14 T cells, whereas expression of CXCR3 was downregulated on KLRG1^+ memory P14 T cells, by repeated rmIFN-γ treatments (Fig. 3e). IFN-γ also induced CD127 downregulation in vitro in cultured memory P14 T cells (Supplementary Fig. 3c), indicating a cell-intrinsic effect of IFN-γ.

Collectively, these data indicate that chronic IFN-γ signals impair maintenance of memory CD8 T cells while promoting their proliferation, even in the presence of CD4 T cells and also that the CD127^high memory subset is more susceptible to IFN-γ-induced attrition than the CD127^low memory subset.

**Ectopic expression of a gain-of-function STAT1 mutant impairs memory CD8 T cell persistence in WT host mice**

The results so far show that continuous IFN-γ signals negatively affect memory CD8 T cell maintenance. We then asked whether this effect of IFN-γ signaling is CD8 T cell-intrinsic. Because IFN-γ receptor-deficient P14 T cells fail to generate functional effector or memory T cells[28], we took a gain-of-function approach by examining the impact of ectopic expression in P14 T cells of wild-type STAT1 (STAT1^WT) or STAT1^R274W, a gain-of-function (GOF) STAT1 mutant associated in human patients with chronic mucocutaneous candidiasis (CMC)[29]. Notably, *Stat1* was upregulated in memory P14 T cells from MHCII^−/− mice (Fig. 2a, Supplementary Table 1), raising the possibility that its up-regulation may contribute to their impaired maintenance.

In vitro-activated Thy1.1 P14 T cells were transduced with empty GFP retrovirus or retrovirus encoding STAT1^WT or STAT1^R274W and transferred into Thy1.2 WT mice that had been infected with Vac-GP33 one day earlier (Fig. 4a). When PBLs were examined on day 8 after infection, 33.2 ± 8.5% of control, 12.1 ± 2.7% of STAT1^WT-transduced, and 11.3 ± 3.6% of STAT1^R274W-transduced P14 cells were GFP^+ (Fig. 4b, Supplementary Fig. 4a). The abundance of KLRG1^lowCD127^high MP and KLRG1^highCD127^low TE in GFP^+ P14 T cells was comparable among the three groups (Fig. 4c), indicating that ectopic expression of STAT1^WT or STAT1^R274W did not perturb development of these effector cell subsets. Since transduction efficiency differed between controls and the other two groups, we normalized the percentage of GFP^+ cells in donor P14 cells at each time point by that of day 8 PBLs and followed changes in the relative percentage of GFP^+ cells over time. The relative percentage of GFP^+ cells in PBLs was less in the STAT1^R274W group than in the empty

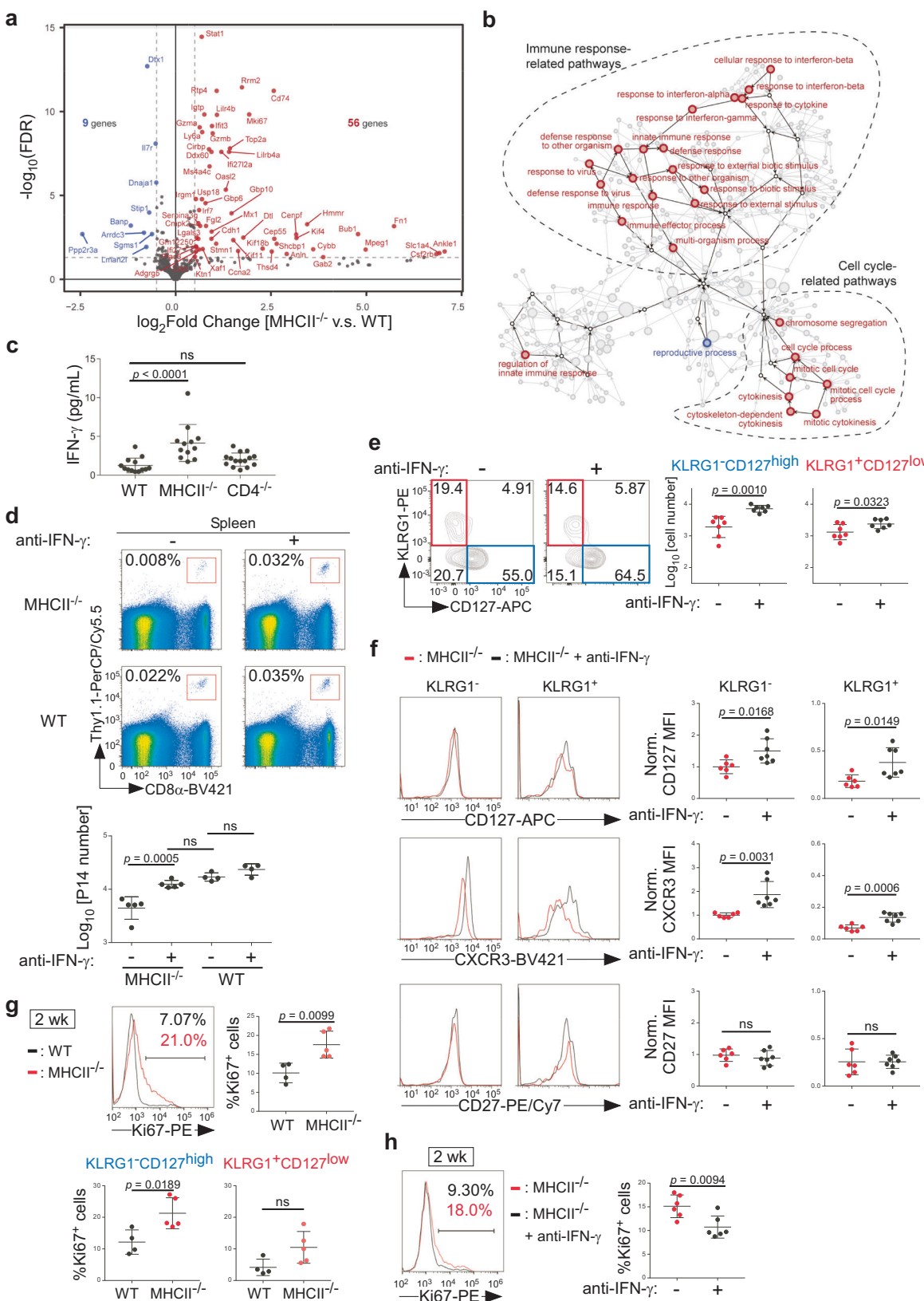

and STAT1[WT] groups at 3, 6 and 9–10 weeks, while the percentage of total P14 cells in PBLs was comparable among the three groups (Fig. 4d, Supplementary Fig. 4a). The reduction in relative representation of STAT1[R274W]-transduced GFP+ cells was also observed in the spleen, liver, and lung when examined 10–11 weeks after infection, while numbers of total P14 cells recovered from these organs were

comparable (Fig. 4d, Supplementary Fig. 4b). These results indicate that STAT1[R274W]-transduced P14 cells show reduced fitness compared to STAT1[WT]-transduced as well as control P14 cells after the effector phase. Of note, although the relative percentage of GFP+ cells was not significantly different in PBLs and livers between the control and STAT1[WT]-transduced groups, it was reduced in spleens and lungs of the

**Fig. 2 | Elevated IFN-γ production is responsible for attrition of CD127$^{high}$ memory CD8 T cells and their increased proliferation in MHCII-deficient mice.** **a**, **b** Thy1.1 memory P14 T cells were generated and transferred into Thy1.2 WT or MHCII$^{-/-}$ secondary host mice, as described in Fig. 1a. Two weeks after the transfer, Thy1.1$^+$CD8α$^+$Vα2$^+$ P14 T cells were sorted from the spleen and subjected to RNA-Seq analysis. **a** Gene expression of P14 T cells from MHCII$^{-/-}$ mice compared with those from WT mice. DEGs (log$_2$[Fold Change] < −0.5 or > 0.5 and FDR < 0.05) are highlighted in red (up in MHCII$^{-/-}$) or blue (down in MHCII$^{-/-}$). **b** GO enrichment analysis of DEGs. Pathways and GO terms significantly enriched in memory P14 T cells from MHCII$^{-/-}$ (red) or WT host mice (blue) are shown ($p < 0.05$). **c** Plasma concentrations of IFN-γ (mean ± SD; WT: $n = 13$; MHCII$^{-/-}$: $n = 12$; CD4$^{-/-}$: $n = 15$). **d**–**h** Thy1.1 memory P14 T cells were generated as in Fig. 1a and transferred into Thy1.2 WT or MHCII$^{-/-}$ mice treated with anti-IFN-γ mAb or PBS. Spleen cells were subjected to flow cytometric analysis 40–50 (**d**–**f**) or 14–15 days (**g**, **h**) after the transfer. **d** Representative flow cytometric profiles of splenocytes. The number of Thy1.1$^+$CD8α$^+$Vα2$^+$ P14 T cells in the spleen is summarized (mean ± SD; $n = 4$ or 5 per group). **e** Representative flow cytometric profiles of Thy1.1$^+$CD8α$^+$Vα2$^+$ P14 T cells

from spleens of MHCII$^{-/-}$ mice treated with anti-IFN-γ or PBS. The number of indicated subsets is summarized (mean ± SD; $n = 7$ per group). **f** Representative histograms of CD127, CXCR3 and CD27 expression on KLRG1$^-$ and KLRG1$^+$ Thy1.1$^+$CD8α$^+$Vα2$^+$ P14 T cells from spleens of MHCII$^{-/-}$ mice treated with anti-IFN-γ (black) or PBS (red). Geometric MFI values of each marker were normalized by the average value of KLRG1$^-$ cells from PBS-treated mice and summarized (mean ± SD; $n = 6$ or 7 per group). **g**, **h** Representative histograms of Ki67 expression in Thy1.1$^+$CD8α$^+$Vα2$^+$ P14 T cells from spleens of MHCII$^{-/-}$ (red) or WT mice (black) (**g**), or of MHCII$^{-/-}$ mice treated with anti-IFN-γ (black) or PBS (red) (**h**). Percentages of Ki67$^+$ cells among total Thy1.1$^+$CD8α$^+$Vα2$^+$ P14 T cells (upper right panel of **g**, **h**) or in the indicated subsets of Thy1.1$^+$CD8α$^+$Vα2$^+$ P14 T cells (lower panels of **g**) are summarized (mean ± SD; $n = 4$ or 5 for **g**, $n = 6$ for **h**). Each symbol represents one mouse. Results are pooled from two independent experiments. Data were analyzed using the Wald test with the Benjamini-Hochberg procedure (**a**), one-way ANOVA with Tukey's multiple comparison test (**c**, **d**) or two-tailed unpaired $t$ test (**e**–**h**). Source data are provided as a Source Data file.

STAT1$^{WT}$ group (Fig. 4d), indicating a partial contribution of STAT1 expression to impaired fitness of P14 T cells. In summary, these results indicate that continuous STAT1 activation can impair maintenance of memory CD8 T cells in a cell-intrinsic manner.

## scRNA-seq reveals proliferating cells resembling TE cells and overall downregulation of gene signatures of long-lived memory cells in MHCII-deficient environments

To gain insights into how MHCII-deficiency impinges on maintenance of memory CD8 T cells, we performed single-cell RNA-seq (scRNA-seq) on memory P14 T cells purified from spleens of MHCII$^{-/-}$ and WT host mice 2 weeks after the secondary transfer, the same time point as the bulk RNA-seq analysis. After filtering and integrating two independent experiments, we retained 11,755 cells from WT host mice and 8813 cells from MHCII$^{-/-}$ host mice. Dimensionality reduction with the uniform manifold approximation and projection (UMAP) and unsupervised clustering yielded 7 clusters (Fig. 5a). Cells of cluster 6 were excluded from analysis because they expressed MHCII-related and B cell transcripts (Supplementary Table 2). Cells of cluster 5 expressed cell cycle-related genes, including *Mki67* and *Top2a*, indicating that they are proliferating cells (Fig. 5b, Supplementary Table 2). Using published bulk RNA-seq datasets for CD127$^{low}$CD62L$^{low}$ (t-T$_{EM}$), CD127$^{high}$CD62L$^{low}$ (T$_{EM}$) and CD127$^{high}$CD62L$^{high}$ (T$_{CM}$) CD8 T cells[14], we annotated cluster 0 as t-T$_{EM}$ cells based on their high-level expression of genes upregulated in t-T$_{EM}$ cells relative to T$_{EM}$ and T$_{CM}$ cells (t-T$_{EM}$ signature), including *Cx3cr1*, *Gzma*, *Klrg1*, *S1pr5*, and *Zeb2* (Fig. 5b, c, Supplementary Table 2, Data 1). Conversely, clusters 1–4 were comprised of both T$_{CM}$ and T$_{EM}$ cells, as they showed high-level expression of genes upregulated in either T$_{CM}$ or T$_{EM}$ cells, relative to t-T$_{EM}$ cells (T$_{CM}$/T$_{EM}$ signature), including *Ccr7*, *Cxcr3*, *Il7r* and *Sell* (Fig. 5b, c, Supplementary Data 1). Notably, clusters 1–4 consisted of cells with high expression levels of genes uniquely upregulated in T$_{CM}$ cells, including *Ccr7* and *Sell*, and cells with low-level expression of such T$_{CM}$ signature genes (Fig. 5b, c, Supplementary Data 1), indicating that T$_{CM}$ and T$_{EM}$ cells constitute a continuum of transcriptionally heterogeneous cell states, rather than discrete subsets.

We then asked how MHCII-deficient environments affect composition and transcriptional states of memory CD8 T cell clusters. The relative abundance of each cluster was not markedly different between the MHCII$^{-/-}$ and WT groups at this time (Fig. 5d). DEGs identified by bulk RNA-seq (Fig. 2a, Supplementary Data 1) were differentially expressed, as expected, in all clusters (Supplementary Fig. 5a). In addition, clusters 0–3 from MHCII$^{-/-}$ mice showed increased expression of genes that belong to the GO term "response to type II IFN", including *Stat1*, compared to those from WT mice, whereas clusters 0–2 and 5 from MHCII$^{-/-}$ mice showed reduced expression of *Il7r* (Fig. 5e, Supplementary Fig. 5b). These results confirm the validity of scRNA-seq results and indicate that most

memory P14 T cells were transcriptionally influenced by increased IFN-γ levels in MHCII$^{-/-}$ mice. On the other hand, genes of "cell cycle process" were upregulated in cluster 5 from MHCII$^{-/-}$ mice (Fig. 5e), a result consistent with enhanced proliferation of P14 T cells in MHCII-deficient environments.

Both self-renewal and long-term survival contribute to maintenance of memory CD8 T cells. We were puzzled by our observations that P14 memory T cells in MHCII$^{-/-}$ host mice exhibited accelerated proliferation, but impaired homeostasis. Since rapid proliferation is a hallmark of TE cells, we hypothesized that proliferating cells in MHCII-deficient environments might adopt a TE cell-like state rather than a self-renewing memory state. Using RNA-seq datasets for TE, MP and naïve CD8 T cells in addition to T$_{CM}$, T$_{EM}$ and t-T$_{EM}$ cells[14], we found that cluster 5 from MHCII$^{-/-}$ mice had higher scores for TE cell signature genes, i.e., genes uniquely upregulated in TE cells relative to all other subsets (Fig. 5f, Supplementary Data 1). This was not simply a consequence of enhanced cell cycling because both cell cycle process-related and -unrelated TE cell signature genes showed higher scores (Supplementary Fig. 5c, Supplementary Data 1). By contrast, all clusters, including cluster 5 from MHCII$^{-/-}$ mice, downregulated genes commonly upregulated in all memory subsets (t-T$_{EM}$, T$_{CM}$ and T$_{EM}$ cells) relative to TE cells (long-lived memory signature), including *Il7r* (Fig. 5f, Supplementary Fig. 5b, Supplementary Data 1). These results indicate that proliferating cells from MHCII$^{-/-}$ mice resemble TE cells rather than long-lived memory cells. Furthermore, of these long-lived memory signature genes, genes of "negative regulation of apoptotic process" were also downregulated in all clusters from MHCII$^{-/-}$ mice (Fig. 5f, Supplementary Data 1), suggesting compromised survival of all memory subsets in MHCII-deficient environments. These results collectively suggest that MHCII-deficient environments impair memory CD8 T cell homeostasis by compromising their long-term survival and by driving proliferating memory cells to undergo terminal differentiation rather than self-renewal.

## Colonic CD8 T cells are the major source of excess IFN-γ in MHCII-deficient mice

Finally, we sought to determine the source of excess IFN-γ in MHCII$^{-/-}$ mice. To this end, we first measured IFN-γ in various tissues. IFN-γ levels were elevated in the colon, but not in the spleen, skin, lung, liver or small intestine of MHCII$^{-/-}$ mice compared to WT mice, whereas IFN-γ levels were not altered in the spleens or colons of CD4$^{-/-}$ mice (Fig. 6a). Next, to identify cell type(s) producing excessive IFN-γ in colons of MHCII$^{-/-}$ mice, we crossed MHCII$^{-/-}$ mice with IFN-γ Venus reporter mice[30]. While the frequency of Venus$^+$ cells was increased in NK1.1$^+$TCRβ$^-$ and CD8β$^+$NK1.1$^-$TCRβ$^+$ cells (Fig. 6b), the number of Venus$^+$CD8β$^+$NK1.1$^-$TCRβ$^+$ cells, but not Venus$^+$NK1.1$^+$TCRβ$^-$ cells, was increased in colons of MHCII$^{-/-}$ mice (Fig. 6c). Although the number of Venus$^+$CD8β$^+$NK1.1$^-$TCRβ$^+$ cells was also increased in spleens of

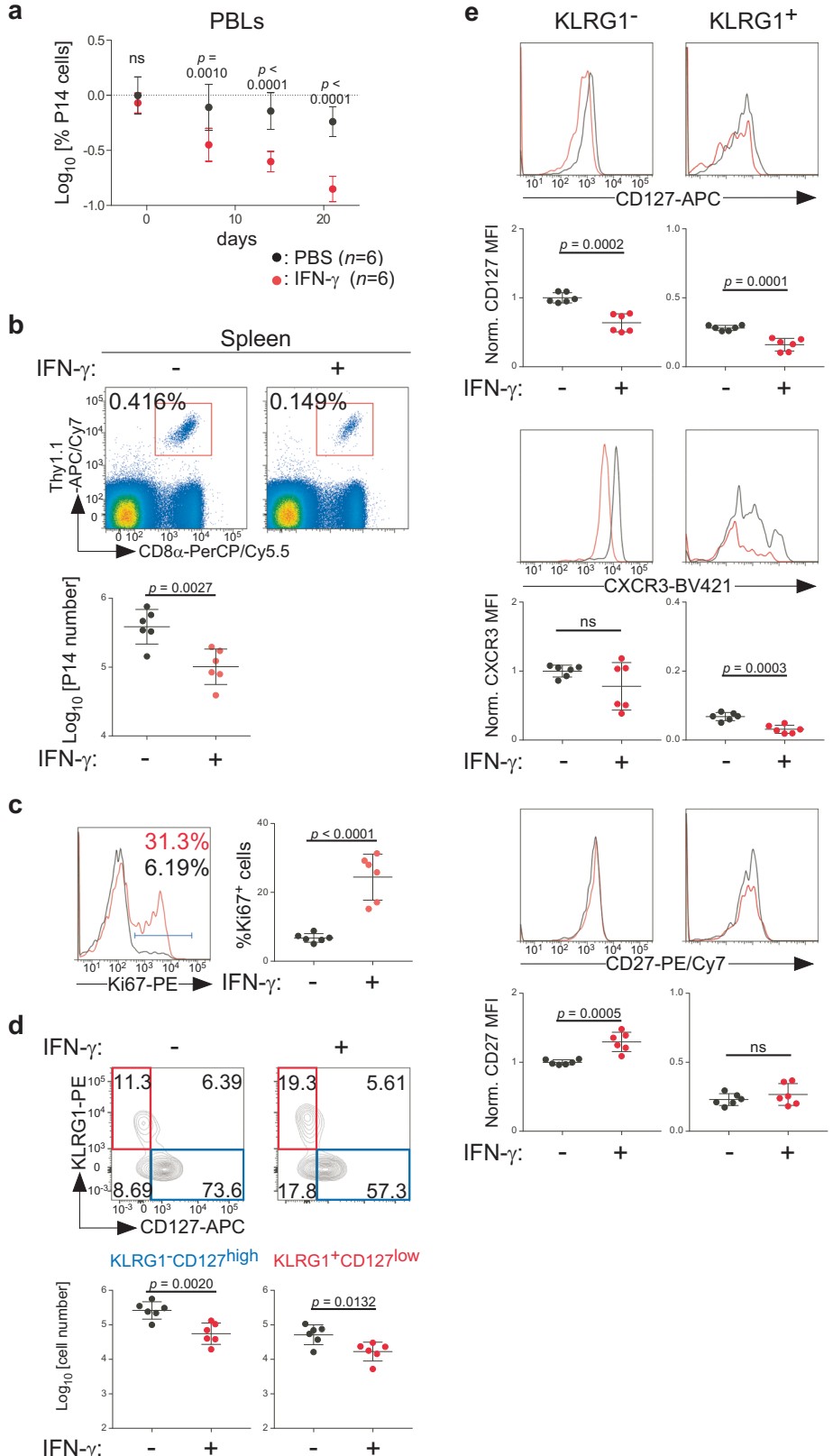

MHCII$^{-/-}$ mice, the number of Venus$^+$CD8β$^-$NK1.1$^-$TCRβ$^+$ cells was reduced. Consequently, the total number of splenic Venus$^+$NK1.1$^-$TCRβ$^+$ cells remained unchanged (Fig. 6c). The number of total CD8β$^+$NK1.1$^-$TCRβ$^+$ cells was also increased in colons, but not in spleens (Supplementary Fig. 6). These results indicate that MHCII-deficiency promotes both expansion of endogenous CD8 T cells and their IFN-γ production in the colon, and suggest that endogenous colonic CD8 T cells are the major source of excessive IFN-γ in MHCII$^{-/-}$ mice.

## Discussion

Mechanisms and factors that regulate memory CD8 T cell homeostasis remain incompletely understood. It has been controversial whether

**Fig. 3 | Repeated IFN-γ administration selectively impairs maintenance of CD127$^{high}$ memory CD8 T cells in the presence of CD4 T cells.** Thy1.1 naïve P14 T cells were transferred into Thy1.2 WT mice and infected with Vac-GP33, as in Fig. 1a. More than 30 days later, rmIFN-γ or PBS was administered to host mice 3 times per week. Splenocytes were subjected to flow cytometric analysis 24−25 days after the initial IFN-γ administration. **a** Time-course of the percentage of Thy1.1$^+$CD8α$^+$Vα2$^+$ P14 T cells in PBLs following the first rmIFN-γ administration (mean ± SD). **b** Representative flow cytometric profiles of splenocytes. The number of Thy1.1$^+$CD8α$^+$Vα2$^+$ P14 T cells in spleens is summarized (mean ± SD). **c** Representative histograms of Ki67 expression in Thy1.1$^+$CD8α$^+$Vα2$^+$ P14 T cells from spleens. The percentage of Ki67$^+$ cells in P14 T cells is summarized

(mean ± SD). **d** Representative flow cytometric profiles of Thy1.1$^+$CD8α$^+$Vα2$^+$ P14 T cells from spleens. The number of indicated subsets is summarized (mean ± SD). **e** Representative histograms of CD127, CXCR3 and CD27 expression on KLRG1$^-$ and KLRG1$^+$ Thy1.1$^+$CD8α$^+$Vα2$^+$ P14 T cells from spleens of WT mice treated with IFN-γ (red) or PBS (black). Geometric MFI values of each marker were normalized by the average value of KLRG1$^-$ cells from PBS-treated mice and summarized (mean ± SD). Each symbol indicates one mouse (*n* = 6 per group). Results are pooled from two independent experiments. Data were analyzed using two-way ANOVA with Sidak's multiple comparisons test (**a**) or two-tailed unpaired *t* test (**b**−**e**). Source data are provided as a Source Data file.

CD4 T cell help constitutes one such factor, since a reduction in the number of adoptively transferred memory CD8 T cells was consistently observed in MHCII-deficient mice, but not in other CD4 T cell-deficient mice[18,19,22,23]. In this study, we showed that elevated IFN-γ production from endogenous colonic CD8 T cells, a previously undescribed phenotype of MHCII-deficient mice, but not CD4 T cell-deficiency, is responsible for impaired maintenance of memory CD8 T cells in MHCII-deficient environments. Chronic IFN-γ signals selectively impair persistence of T$_{CM}$ and/or T$_{EM}$ cells relative to t-T$_{EM}$ cells/LLECs, a recently described memory subset that displays characteristics of both memory cells and terminal effector cells. Mechanistically, MHCII-deficient environments exhaust memory CD8 T cells, probably by impairing their long-term survival and by skewing proliferating CD127$^{high}$ memory cells toward terminal differentiation rather than self-renewal.

Although we did not find any impact of CD4 T cell-deficiency on the number of memory CD8 T cells, it contributed to downregulation of CD127 and CXCR3 on memory CD8 T cells. While a previous study reported a modest reduction in the number of memory CD8 T cells in CD4$^{-/-}$ host mice[24], another study found no effect of CD4-deficiency on the number and phenotype of memory CD8 T cells[23]. Although the root of these apparently contradictory observations is unclear, they may well be explained by differences in experimental conditions. Thus, it remains possible that under certain conditions, CD4 T cell-deficiency may impair memory CD8 T cell homeostasis, perhaps through downregulation of memory cell-associated molecules such as CD127, IL-7Rα, which are essential for their survival.

How do chronic IFN-γ signals compromise maintenance of memory CD8 T cells, particularly the CD127$^{high}$ subset, in MHCII-deficient environments? Importantly and counterintuitively, anti-IFN-γ treatments increased their numbers while inhibiting their proliferation in MHCII$^{-/-}$ host mice, whereas IFN-γ administration decreased their numbers while accelerating their proliferation in WT host mice, suggesting that IFN-γ-dependent excessive proliferation is mechanistically linked to impaired memory CD8 T cell maintenance. Our scRNA-seq analysis may explain how increased proliferation leads to impaired memory CD8 T cell maintenance. We found that while proliferating cells expressed long-lived memory cell gene signatures in WT host mice, they downregulated these signature genes and upregulated TE cell gene signatures in MHCII$^{-/-}$ host mice. These results suggest that while CD127$^{high}$ memory cells adopt a self-renewing state upon proliferation in MHCII-sufficient environments, they adopt a short-lived TE cell-like state and subsequently undergo cell death in MHCII-deficient environments. In addition, our scRNA-seq analysis also revealed downregulation of long-lived memory signature genes, particularly those related to negative regulation of apoptosis in all memory cell clusters in MHCII-deficient environments, suggesting impaired survival. In conclusion, our results suggest that chronic IFN-γ signaling compromises memory CD8 T cell homeostasis both by impairing the long-term survival of all memory CD8 T cell subsets and by redirecting self-renewing CD127$^{high}$ memory cells toward terminal differentiation and subsequent cell death.

A previous study has reported that IFN-γ deficiency favors the differentiation of antigen-primed CD8 T cells into MP rather than TE cells during the effector phase[31], although it was unclear whether IFN-γ acts directly on CD8 T cells. These findings are consistent with our present results and collectively suggest that IFN-γ/STAT1 signaling selectively targets CD127$^{high}$ MP or memory cells relative to CD127$^{low}$ TE or t-T$_{EM}$ cells/LLECs, driving terminal differentiation while inhibiting self-renewal and long-term survival. Further studies are needed to clarify the underlying molecular mechanisms. It has been suggested that in human CMC patients, STAT1 GOF mutants inhibit development of protective Th17 responses by interfering with STAT3 activity[32]. Since STAT3 is required for development and maintenance of memory CD8 T cells[33,34], STAT1 activation by IFN-γ may inhibit their development and maintenance in part by interfering with STAT3 activity.

We have shown that MHCII molecules negatively regulate IFN-γ production in colonic CD8 T cells in the absence of CD4 T cells. Further studies are needed to identify molecular and cellular mechanisms. To this end, it is necessary to investigate whether MHCII molecules directly limit IFN-γ production in colonic CD8 T cells by engaging the inhibitory MHCII receptor, LAG-3, expressed on these cells, and/or do so indirectly by transducing reverse signals in MHCII-expressing antigen-presenting cells to suppress their stimulatory functions or by selecting and activating Foxp3$^+$ regulatory T cells. Since some Foxp3$^+$ T cells with suppressive activity develop in CD4-deficient mice[35], they may limit IFN-γ production in colonic CD8$^+$ T cells in these mice. Although some Foxp3$^+$ T cells also develop in MHCII-deficient mice[36], they may be quantitatively or qualitatively different from those in CD4-deficient mice and may be unable to limit IFN-γ production in colonic CD8 T cells. In addition, it is important to determine the role of colonic microbiota in excessive IFN-γ production, which may explain colon-specific dysregulation of IFN-γ production.

It also remains to be determined whether and to what extent chronic IFN-γ signaling contributes to attrition of memory CD8 T cells in more physiologically relevant settings, such as persistent infection and chronic inflammation. Nevertheless, our findings may have important implications in understanding and manipulating immunological memory in humans. First, human CMC patients bearing autosomal dominant STAT GOF mutations, including STAT1$^{R274W}$, often suffer from recurrent bacterial or viral infections, including herpesvirus infections, control of which depends on memory CD8 T cells, but the mechanisms remain poorly understood[37]. In light of our present findings, it is tempting to speculate that these patients suffer such infections because of defective maintenance of memory CD8 T cells. A recent report showed that STAT1$^{R274W}$ knock-in mice exhibit increased susceptibility to γ-herpesvirus 68[38]. However, it remains to be investigated whether and how this mutation affects differentiation and maintenance of memory CD8 T cells. Second, there is epidemiological and experimental evidence that chronic infections impair immune responses to unrelated pathogens and vaccines[39,40]. Previous studies using a chronic viral infection model have shown that chronic inflammation reduces the number of bystander memory CD8 T cells and skews them toward the KLRG1$^+$CD127$^{low}$ phenotype[25,41]. While the IL-6/STAT1 pathway contributes to this phenotypic skewing, IL-6 alone

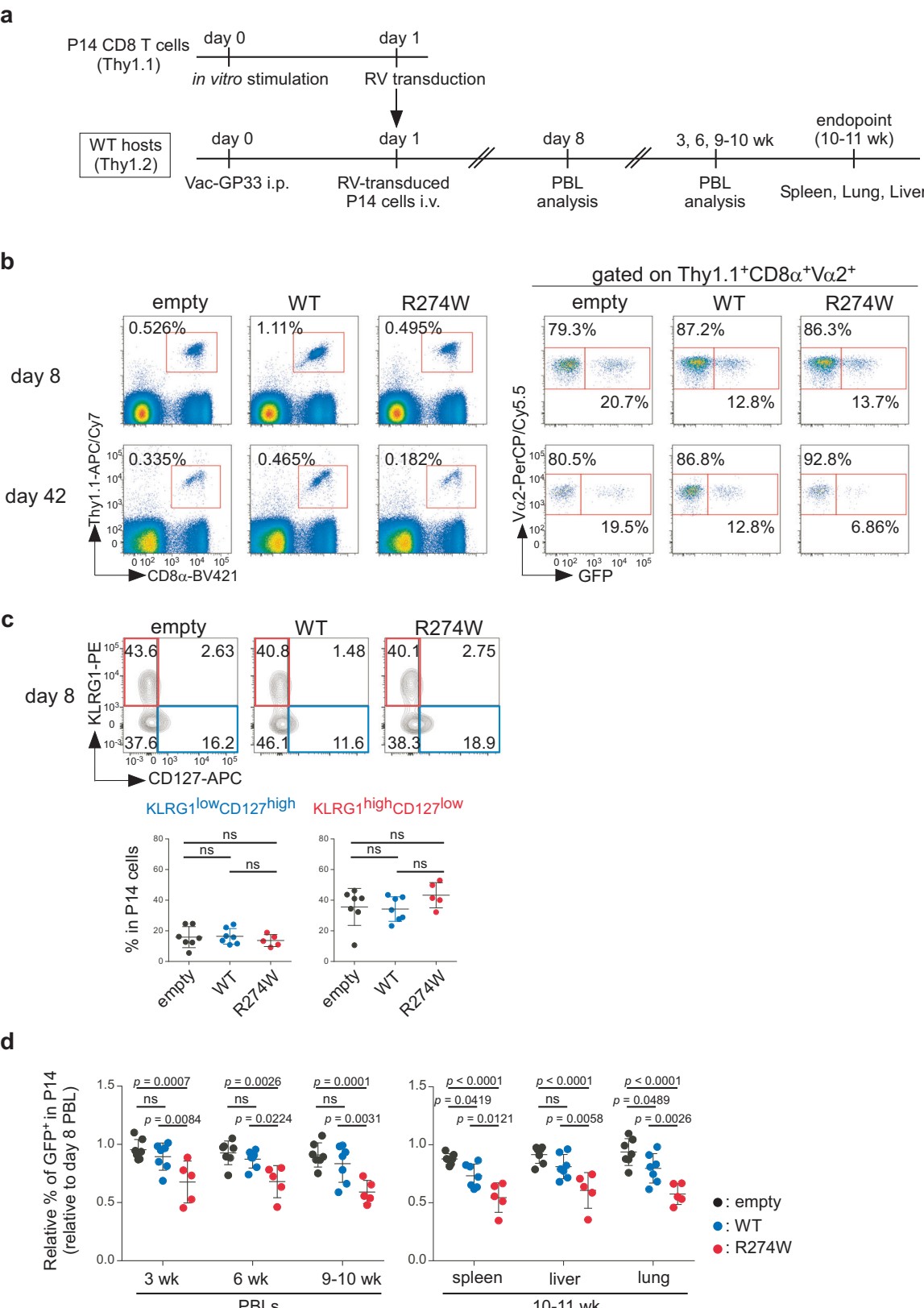

failed to reduce the number of bystander memory CD8 T cells[41]. In contrast, Dudani et al. reported that attrition of bystander memory CD8 T cells that occurs during infection with heterologous pathogens is abrogated by IFN-γ neutralization, suggesting an important role for the IFN-γ/STAT1 pathway[42]. However, it remains unclear whether IFN-γ signaling impairs their maintenance in a cell-intrinsic manner. Future

studies using inducible ablation of IFN-γ receptor or STAT1 in memory CD8 T cells will allow us to address the cell-intrinsic role of chronic IFN-γ/STAT1 signaling in attrition of memory CD8 T cells during persistent infections and chronic inflammation. If proven, our findings may be relevant for vaccination of patients with chronic inflammatory diseases or people in countries where infectious diseases are prevalent. Future

**Fig. 4 | Ectopic expression of a gain-of-function STAT1 mutant impairs persistence of memory CD8 T cells in WT host mice. a** Experimental scheme. Thy1.1 naïve P14 T cells were activated and transduced in vitro with pMCs-IRES-GFP retrovirus (RV) encoding WT STAT1 or STAT1 R274W mutant, or empty RV, and transferred into Thy1.2 WT host mice ($4–5 \times 10^5$ cells per mouse) which had been infected with Vac-GP33 one day earlier. PBLs were collected and analyzed by flow cytometry 8 days, 3 weeks, 6 weeks, and 9–10 weeks after the infection. Lymphocytes from spleens, livers, and lungs were harvested and subjected to flow cytometric analysis 10–11 weeks after infection. **b** Representative flow cytometric profiles of total PBLs (left) and Thy1.1$^+$CD8α$^+$Vα2$^+$ PBLs (right) on day 8 (upper) or

42 (lower) after the infection. **c** Representative flow cytometric profiles of Thy1.1$^+$CD8α$^+$Vα2$^+$GFP$^+$ P14 T cells in PBLs on day 8 after infection. Percentages of indicated subsets in effector P14 T cells are summarized (mean ± SD). **d** The percentage of GFP$^+$ cells in Thy1.1$^+$CD8α$^+$Vα2$^+$ P14 T cells relative to that of GFP$^+$ cells in day 8 PBLs (mean ± SD). Percentages for indicated times (left) or organs (right) were divided by the percentage of day 8 PBLs. Each symbol indicates one mouse (empty: $n = 7$; WT: $n = 7$; R274W: $n = 5$). Results are pooled from two independent experiments. Data were analyzed using one-way ANOVA with Tukey's multiple comparisons test (**c**) or two-way ANOVA with Tukey's multiple comparisons test (**d**). Source data are provided as a Source Data file.

studies should address whether and how chronic IFN-γ signaling impairs maintenance of memory CD8 T cells and how IFN-γ suppression could be employed therapeutically to develop long-lasting immunological memory in humans.

## Methods

### Mice

C57BL/6 J, B6.PL-*Thy1*$^a$/CyJ (stock number: 000406), B6.129S2-*Cd4*$^{tm1Mak}$/J (CD4$^{-/-}$) (stock number: 002663), and B6.129S2-*H2*$^{dlAb1–Ea}$/J (MHCII$^{-/-}$) (stock number: 003584) mice were obtained from the Jackson Laboratory. P14 TCR transgenic mice on a C57BL/6 J background were kindly provided by Dr. Takashi Saito with permission of Dr. Hanspeter Pircher and crossed with B6.PL-*Thy1*$^a$/CyJ mice. B6.IFN-γ Venus BAC transgenic reporter mice[30] were crossed with MHCII$^{-/-}$ mice. Mice were kept under specific pathogen-free conditions. All animal experiments were performed according to protocols approved by the Institutional Animal Care and Use Committee of RIKEN Yokohama Branch and by the Animal Care and Use Committee of The University of Tokyo.

### Adoptive transfer, viral infection, and antibody or cytokine treatments

Single-cell suspensions were prepared from spleens and peripheral lymph nodes (cervical, axillar, brachial, and inguinal) of Thy1.1 P14 TCR transgenic mice. Thy1.1 P14 CD8 T cells were purified using a mouse CD8a$^+$ T Cell Isolation Kit (Miltenyi Biotech) and transferred intravenously into Thy1.2 B6 mice ($2–5 \times 10^4$ cells per mouse). Vaccinia virus expressing GP$_{33–41}$ (Vac-GP33) was propagated in and titrated with 143B osteosarcoma cells, and the virus stock was prepared[43]. Mice that had received P14 CD8 T cells were infected intraperitoneally with $3–5 \times 10^6$ PFU of Vac-GP33 for immunization. Thirty to fifty days after infection, CD8 T cells containing Thy1.1 P14 T cells were purified from spleens of infected B6 mice using a mouse CD8a$^+$ T Cell Isolation Kit (Miltenyi Biotech) and transferred intravenously into secondary host mice ($3–5 \times 10^5$ P14 cells per host).

For depletion of CD4 T cells, mice were injected intravenously with 200 μg of anti-CD4 mAb (GK1.5) (ATCC, prepared in-house) 7 or 6 and 3 days before transfer of memory P14 T cells, and then treated with 100 μg of anti-CD4 mAb on day 4 and every seven days thereafter. For neutralization of IFN-γ, WT and MHCII$^{-/-}$ mice were injected intravenously with 200 μg of anti-IFN-γ mAb (XMG1.2) (Bio X Cell) on days 7 and 3 before the transfer of memory P14 T cells, and then injected with 100 μg of anti-IFN-γ mAb on day 3 and every seven days thereafter. For IFN-γ treatments, WT mice that had been inoculated with naïve P14 T cells and infected Vac-GP33 were injected intravenously with 10 μg of recombinant murine IFN-γ (Biolegend) every 2 or 3 days (3 times per week), starting from >30 days post-infection.

### Cell isolation, flow cytometry, and cell sorting

Single-cell suspensions from spleens and lymph nodes were obtained by gently forcing tissues through a nylon mesh grid in PBS containing 2% FBS. Splenic and blood erythrocytes were eliminated with ACK lysis buffer (Lonza). To analyze expression of KLRG1 and memory T cell-

associated markers (CD27, CD127, CXCR3), spleen cells were first depleted of B cells and adherent cells by panning before staining. Mononuclear leukocytes from livers and lungs were prepared as follows. After perfusion with PBS, lungs were cut into small pieces and incubated for 80 min at 37 °C with occasional vortexing in PBS containing 2% FBS, 100 U/mL collagenase type I (Worthington) and 100 μg/mL DNase I (Roche). Livers were perfused, minced by forcing the organ through a nylon mesh grid, and digested in PBS containing 100 U/mL collagenase type I (Worthington) and 100 μg/mL Dnase I (Roche) for 45 min at 37 °C. The resulting lung and liver suspensions were passed through a 70-μm cell strainer, resuspended in 40% Percoll (GE Heathcare Life Sciences) layered on 60% Percoll, and centrifuged at $1000 \times g$ for 20 min. Leukocytes were recovered from the interphase. To obtain colonic lamina propria leukocytes, colons were cut into 1-cm pieces, washed with PBS, and incubated twice with stirring for 20 min at 37 °C in HBSS containing 10 mM HEPES and 5 mM EDTA to remove epithelial cells. The remaining tissues were cut into small pieces, digested twice with stirring for 30 min at 37 °C in HBSS(+) containing 1% BSA, 200 U/ml collagenase type 4 and 100 μg /ml DNase I, vortexed and passed through a 70-μm cell strainer. Leukocytes were then isolated on a 40/80% Percoll density gradient.

Leukocytes were washed and stained with appropriate mAbs for 25 min after incubation with anti-FcγR mAb (2.4G2, prepared in-house) for 5 min on ice. Dead cells were stained with SYTOX blue (Invitrogen). A representative gating strategy is shown in Supplementary Fig. 7. To detect Ki67, cells were stained for cell surface markers, and then dead cells were stained with Fixable Viability Dye eFluor 780 (eBioscience). Cells were fixed and permeabilized using a Foxp3 Staining Buffer Set (eBioscience) according to the manufacturer's instructions. Cells were analyzed on FACSCanto II (BD Biosciences) or CytoFLEX S (Beckman Coulter) flow cytometers and analyzed with FlowJo software (TreeStar).

For bulk RNA-seq and scRNA-seq of memory P14 T cells, Thy1.1$^+$CD8α$^+$Vα2$^+$ cells were sorted from spleens using FACSAria II or III (BD Biosciences) after magnetic enrichment of Thy1.1$^+$ cells using anti-APC beads and MACS LS columns (all Miltenyi). Purity was always >98%.

For in vitro culture of memory P14 T cells, Thy1.1$^+$CD8α$^+$ cells were purified from spleens, cultured with or without 10 μg/mL rmIFN-γ (Biolegend) in the presence of IL-15 (1 ng/mL, Biolegend) for 7 days, and analyzed by flow cytometry.

### Cytokine measurements

Blood was collected in BD Microtainer Blood Collection Tubes (BD Biosciences) and centrifuged at $2000 \times g$ for 3 min. Supernatants were collected as plasma and stored frozen until use. Spleens were minced by gently forcing them through a nylon mesh grid in PBS and the cell suspension was centrifuged at $500 \times g$ for 6 min to collect the supernatant as splenic interstitial fluid, which was stored frozen until use. Plasma concentrations of IFN-γ were determined by validated electrochemiluminescence V-PLEX immunoassay, following the manufacturer's instructions (Meso Scale Discovery). The standard range was 0.19–785 pg/mL, while the lower limit of detection was 0.04 pg/mL. Concentrations of other cytokines in plasma or splenic interstitial fluid

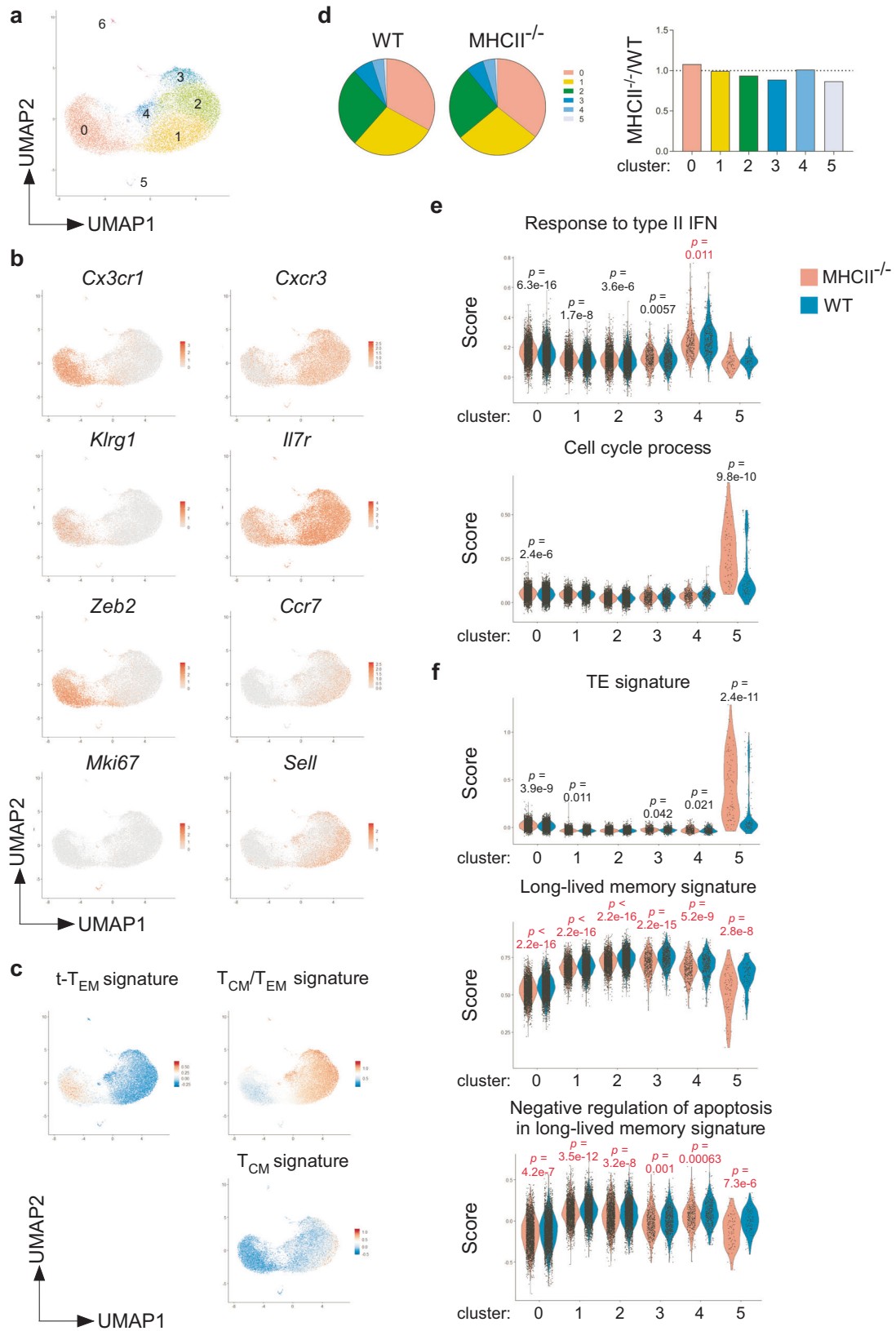

were determined using ELISA kits (IFN-β Biolegend; IL-15/15 R complex: eBioscience; IL-7: R&D) according to the manufacturer's instructions. The standard range was 7.8–500, 31.2–2000, and 7.8–500 pg/mL, while the lower limit of detection was 1.9, 8.3, and 4 pg/mL, for IFN-β, IL-7, and IL-15/15 R complex, respectively. To measure IFN-γ concentrations, all tissues except skin were placed in PBS supplemented

with protease inhibitors (a Complete Mini tablet, Roche) and then homogenized using Micro Smash (TOMY, MS-100R) at 4000 rpm for 60 sec at 4 °C. For skin, NP-40 cell lysis buffer was used instead of PBS. Supernatants were collected and protein concentrations in the supernatants were determined using a Protein Assay BCA kit (Nacalai). Concentrations of IFN-γ in the supernatants were determined using an

**Fig. 5 | scRNA-seq on memory CD8 T cells reveals proliferating cells resembling TE cells and global downregulation of gene signatures of long-lived memory cells in MHCII$^{-/-}$ host mice.** Thy1.1$^+$CD8α$^+$Vα2$^+$ memory P14 T cells were sorted from spleens of WT and MHCII$^{-/-}$ host mice 2 weeks after the secondary transfer, as in Fig. 2a and subjected to scRNA-seq analysis. **a** Integrated UMAP plot of scRNA-seq data (MHCII$^{+/+}$: $n = 2$; MHCII$^{-/-}$: $n = 2$). A total of 7 clusters (cluster 0–6) were identified. For all UMAP plots, each dot is a single cell. **b** Relative expression levels of indicated genes. **c** Relative enrichment of genes that are uniquely upregulated in t-T$_{EM}$ cells relative to T$_{CM}$ and T$_{EM}$ cells (t-T$_{EM}$ signature), those upregulated in T$_{CM}$ or T$_{EM}$ cells relative to t-T$_{EM}$ cells (T$_{CM}$/T$_{EM}$ signature), and those uniquely upregulated in T$_{CM}$ relative to T$_{EM}$ and t-T$_{EM}$ cells (T$_{CM}$ signature). **d** The percentage of each cluster in P14 T cells from indicated host mice (left). Ratio of the percentage of

each cluster in the MHCII$^{-/-}$ group to that in the WT group is shown (right). **e** Relative expression levels of genes belonging to the GO terms "response to type II IFN" (GO:0034341) and "cell cycle process" (GO:0022402). For all violin plots, each dot is a single cell. **f** Relative expression levels of genes uniquely upregulated in TE cells relative to naïve, MP, T$_{CM}$, T$_{EM}$ and t-T$_{EM}$ CD8 T cells (TE signature), those commonly upregulated in T$_{CM}$, T$_{EM}$ and t-T$_{EM}$ cells relative to TE cells (long-lived memory signature) and long-lived memory signature genes that belong to the GO term "negative regulation of apoptotic process" (GO:0043066). For all violin plots, each dot is a single cell. *P*-values were calculated using the Wilcoxon rank-sum test. When the average value of the MHCII$^{-/-}$ group is higher or lower than that of the WT group, the *p*-value is indicated in black or red characters, respectively. Source data are provided as a Source Data file.

ELISA Max™ Deluxe Set Mouse IFN-γ (BioLegend), with a standard range of 7.8–500 and a lower limit of detection of 4 pg/mL, according to the manufacturer's instructions.

### Quantitative RT-PCR
Total RNA was extracted from spleens using Isogen (Nippon Gene). Total RNA was reverse transcribed using a Superscript VILO cDNA synthesis kit (Invitrogen). Real-time PCR was performed using THUNDERBIRD SYBR qPCR mix (Toyobo) on an ABI/PRISM 7000 sequence detection system (Applied Biosystems). Primer sequences were as follows. *Il7*: 5′-TCCTCCACTGATCCTTGTTC-3′ and 5′-CTTCAACTTGC-GAGCAGCAC-3′; *Il15*: 5′-GTGACTTTCATCCCAGTTGC-3′ and 5′-TTCCTTGCAGCCAGATTCTG-3′; *Hprt*: 5′-TCCTCCTCAGACCGCTTTTT-3′ and 5′-CCTGGTTCATCATCGCTAATC-3′.

### Retroviral constructs and transduction
Mouse STAT1 cDNA was amplified from the FANTOM 3 clone D430002020[44] and cloned into the pMCs-IRES-GFP vector[45]. The STAT1$^{R274W}$ mutant was generated using a KOD -Plus- Mutagenesis Kit (Toyobo) and mouse STAT1 cDNA as a template and cloned into pMCs-IRES-GFP vectors. Plat-E packaging cells[46] were cultured in DMEM, high glucose (Sigma) supplemented with 10% FBS and 10 mM HEPES and transfected with retroviral vectors using FuGENE 6 Transfection Reagent (Promega). Retroviral supernatants were collected 48 h later. CD8 T cells from naïve P14 TCR transgenic mice were activated with plate-bound anti-CD3ε mAb (clone 145-2C11; coated at 1 μg/mL, Biolegend), 0.5 μg/mL anti-CD28 mAb (clone 37.51; Biolegend), and 10 ng/mL recombinant mouse IL-2 (R&D). Viral transduction was performed as follows[47]; 24 h after stimulation, Thy1.1 P14 T cells were infected using viral supernatant in the presence of 5 μg/mL polybrene and centrifuged at 2000 × *g* for 60 min at 32 °C. Three to four hours after spin infection, cells were collected and transferred into Thy1.2 B6 mice (4–5 × 10$^5$ per mouse) that had been infected with Vac-GP33 one day earlier.

### Bulk RNA-sequencing
Splenic Thy1.1$^+$CD8α$^+$Vα2$^+$ P14 T cells (1–3 × 10$^4$ per sample) were sorted from MHCII$^{-/-}$ and WT secondary host mice in duplicate. Total RNA was purified with TRIzol Reagent (Invitrogen) and then with an RNeasy mini kit (Qiagen). cDNA was synthesized, amplified, purified, and used for library preparation with a SMARTer Ultra Low Input RNA Kit for Sequencing-v3 (Clontech). Libraries were prepared and sequenced with 100-bp paired-end reads on a HiSeq 2500 (Illumina) by Takara Inc. Reads were aligned to the mouse reference genome (UCSC mm9). The DESeq2 R-package was used to perform differential gene expression analysis. DEGs were defined as genes that had log2 fold-changes of >0.5 or <−0.5 and false discovery rate (FDR) of <0.05. Enrichment analysis based on Gene Ontology (GO) was performed using the GAGE algorithm[48], and the FDR was calculated with the Benjamini-Hochberg procedure from the *p*-value for multiple testing. Enrichment results (Fig. 2b) were visualized using an in-house algorithm based on d3.js.

### Single-cell RNA-sequencing
Single-cell RNA-seq of FACS-sorted memory P14 T cells was performed on a Chromium instrument (10X Genomics). Cells were washed and resuspended in PBS supplemented with 0.04% BSA. Approximately 6–9 × 10$^4$ sorted P14 T cells were loaded and partitioned into Gel Bead In-Emulsions (GEMs). Single-cell libraries were prepared using Chromium Next GEM Single Cell 3′ Reagent Kits v3.1 (10X Genomics) according to the manufacturer's instructions and sequenced on an Illumina NovaSeq 6000 system to generate 50-bp paired-end reads. Alignment, filtering, barcode counting, and molecular identifier counting were done using Cell Ranger v.5.0.1. Further data analysis was performed using Seurat v.4.0.1[49]. Low-quality cells, determined by percent mitochondria <5, nFeature_RNA < 1000 or >5000, were removed. Counters were normalized using the LogNormalize function. scRNA-seq datasets of two independent experiments were integrated using "Harmony" in Seurat. Dimensionality reduction and cluster identification were performed with UMAP (dims = 1:30), FindNeighbors (dims = 1:30), and FindClusters (resolution = 0.3). Differentially expressed genes for each cluster were determined using the FindAllMarkers function (min = 0.1 and logfc.threshold = 0.1). Gene signatures of t-T$_{EM}$, T$_{EM}$, T$_{CM}$, and TE cells were selected as differentially expressed genes among different subsets by DESeq2 (log2 fold-change >1 or <−1 and adjusted *p*-value < 0.05) using a published bulk RNA-seq dataset (GSE157072). To calculate gene signature scores among single cells, we used the AddModuleScore function in the Seurat R package using various gene sets (Supplementary Data 1). Statistical significance was determined with Wilcoxon tests using the ggpubr package.

### Statistics and reproducibility
Data analysis was performed using Prism 5 or 7 (GraphPad Software, Inc.), except for bulk RNA-seq and scRNA-seq experiments. Statistical significance was evaluated by unpaired *t* test, one-way ANOVA with Tukey's multiple comparisons test, or two-way ANOVA with Sidak's or Tukey's multiple comparisons test, as indicated in the figure legends. *P*-values are shown in the graphs when the differences between groups are *p* ≤ 0.05, while ns, not significant, is shown when *p*-values are >0.05. In Supplementary Figs., *p*-values are represented as follows: *$p$ ≤ 0.05; **$p$ ≤ 0.01; ***$p$ ≤ 0.001; ****$p$ ≤ 0.0001. *n* represents number of biological replicates. For all experiments, at least three biological replicates were analyzed in at least two independent experiments. The number of biological replicates is indicated in each figure legend. No data were excluded from the analysis. Experiments were not randomized; animals were assigned to groups based on genotype or treatment. Sample sizes were chosen according to previous publications (e.g., ref. 22).

### Reporting summary
Further information on research design is available in the Nature Portfolio Reporting Summary linked to this article.

## Data availability
RNA-seq and scRNA-seq data are available in the Genomic Expression Archive (GEA) of the DNA Data Bank Japan (DDBJ) under the accession

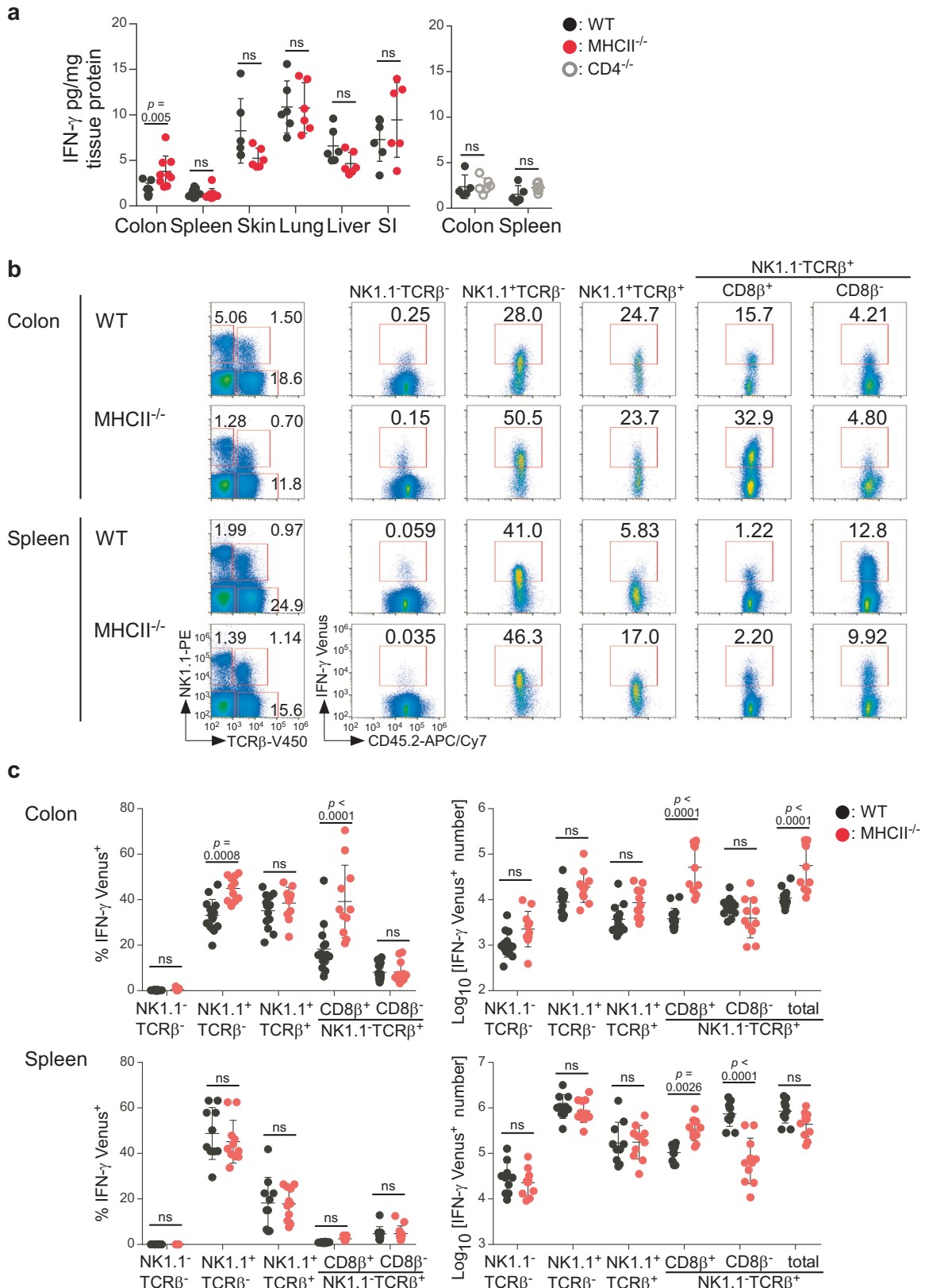

**Fig. 6 | Excess IFN-γ is produced primarily by colonic CD8 T cells in MHCII-deficient mice. a** IFN-γ levels in different tissues of WT, MHCII⁻/⁻, and CD4⁻/⁻ mice (*n* = 5–9 per group). IFN-γ levels were normalized by total protein levels. SI; small intestine. **b** Flow cytometric analysis of colonic and splenic lymphocytes from IFN-γ Venus MHCII⁻/⁻ and MHCII⁺/⁻ (WT) mice. Representative flow cytometric profiles of

IFN-γ Venus expression levels in the indicated subsets are shown. **c** Frequency and number of IFN-γ Venus⁺ cells in the indicated subsets are shown (mean ± SD; MHCII⁻/⁻: *n* = 11; WT: *n* = 10). Each symbol represents one mouse. Data were analyzed using multiple two-tailed unpaired *t* tests (**a**) or two-way ANOVA with Sidak's multiple comparisons test (**c**). Source data are provided as a Source Data file.

numbers; E-GEAD-723 E-GEAD-724. Source data are provided with this paper.

## Code availability

Source codes for bulk RNA-seq and scRNA-seq analysis are available at: https://github.com/eiryo-kawakami/Setoguchi_Tcell_2024, https://github.com/RukaSetoguchi/MHC_class_II.

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

## Acknowledgements

We thank Hiroyuki Yoshitomi and Ryuichi Murakami for technical support of scRNA-seq data analysis, Takako Kato and Takahisa Miyao for technical assistance, Makoto Kurachi for Vac-GP33, Michio Tomura for the 143B cell line, Piero Carninci for the murine STAT1 FANTOM 3 clone, and Toshio Kitamura for the pMC-IRES-GFP retroviral vector and the Plat-E packaging cell line. This work was supported in part by MEXT KAKENHI (23790549, 26860325, 18K07051 to R.S.), the Japan Initiative for World-leading Vaccine Research and Development Centers from the Japan Agency for Medical Research and Development under Grant Number JP223fa627001 (to S.H.), and Kato Memorial Bioscience Foundation (R.S.). scRNA-seq analysis was supported by JSPS KAKENHI Grant Number 16H06279 (PAGS) (to R.S.).

## Author contributions

R.S. designed the study and performed experiments. M.K. provided IFN-γ Venus reporter mice, and T.S. performed experiments with these mice. H.K. constructed the STAT1^R274W retrovirus vector. E.K. analyzed bulk RNA-seq data. R.S. and S.H. analyzed the data and wrote the manuscript. S.H. and T.Y. provided financial support.

## Competing interests

The authors declare no competing interests.
