## [Peer Review File · Nature Communications]

Memory CD8 T cells are vulnerable to chronic interferon- γ signals but not to CD4 T cell deficiency in MHCII-deficient miceEditorial Note: This manuscript has been previously reviewed at another journal that is not operating a transparent peer review scheme. This document only contains reviewer comments and rebuttal letters for versions considered at *Nature Communications*.

REVIEWERS' COMMENTS

Reviewer #1 (Remarks to the Author):

This work resolves an important outstanding issue on the dependence of the maintenance of memory CD8 T cells on CD4 T cells. As stated previously, the paper convincingly shows that CD8 T cell memory maintenance defects are not the result of loss of CD4 T cells in MHC class II deficient mice, but the cause of elevated IFN γ levels in these mice. These findings show that memory CD8 T cell maintenance does not require CD4 T cells. This conclusion represents an important message for the scientific community that is both clearly and convincingly presented in the manuscript. The authors have satisfactorily addressed my concerns on the previous version of the manuscript. It is unfortunate that the authors were unable in the short-term to resolve the points that I raised on showing the relevance of IFN γ -driven attrition in a chronic model and the impact of IFN γ on the maintenance tissue-resident memory T cells. However, the authors have carefully explained how they would need to experimentally address these points and this will likely take them substantial time. I do not think it is necessary to delay the current paper to wait for the additional datasets. It is very nice to see that they have been able to trace the source of IFN γ in the MHC class II deficient mice to CD8 T cells in the colon. The new dataset is informative, convincing and adequately addresses the concern that I raised on this point. My final point on the scRNAseq dataset has also been carefully addressed by the authors. They have convincingly argued that the kinetics of the memory attrition process do not permit large differences to appear in the single cell transcriptomics data. It is also very helpful that this point is now better supported in the revised discussion of the manuscript. Furthermore, they have been able to provide further support for the involvement of enhanced proliferation in this process using their new Ki67 labeling dataset. Taken together, I find that the message of this paper warrants publication in its current form and I do not have any further concerns that require attention.

Reviewer #2 (Remarks to the Author):

This manuscript is much improved due in particular to two key additional data sets that have allowed for a more advanced interpretation of the results.

Firstly, analysis of proliferation of these cells has enabled the interpretation that IFN γ is driving differentiation rather than steady state survival and homeostatic turnover in Tcm cells. This makes sense of early data indicating that the IFN γ preferentially targets less differentiated memory T cells. The study should cite early work by John Harty's group showing that inflammatory signals reduce the establishment of CD8 T cell memory [PMID: 17579021].

Secondly, analysis of the source of the IFN γ (colonic CD8 T cells predominantly) enables some speculation as to how loss of MHCII might drive this. The authors include some limited discussion of this, however there should be some discussion around regulatory T cells. Instinctively, one imagines that production of the proinflammatory cytokine IFN γ in MHCII $^{-/-}$ mice might occur due to a lack of regulation. There are no Tregs in MHCII $^{-/-}$ mice. Are there any in CD4 $^{-/-}$ mice?

This is an excellent paper with important results. My only other suggestion would be to alter

the title to reflect the more significant finding that persistence of CD8 T cell memory is not dependent on CD4 T cells, but rather on low levels of IFN γ . Or that loss of CD8 T cells in MHCII^{-/-} mice is not due to dependence on CD4 T cells, but to high IFN γ levels. As it stands the title states something that we've known for some time (i.e. that IFN γ can erode CD8 T cell memory).

A point-by-point response to reviewers' suggestions

We sincerely thank both reviewers for their positive comments and for taking the time to review our manuscript. We have revised our manuscript in response to reviewer #2's comments.

REVIEWERS' COMMENTS

Reviewer #2 (Remarks to the Author):

This manuscript is much improved due in particular to two key additional data sets that have allowed for a more advanced interpretation of the results.

Firstly, analysis of proliferation of these cells has enabled the interpretation that IFN γ is driving differentiation rather than steady state survival and homeostatic turnover in T_{cm} cells. This makes sense of early data indicating that the IFN γ preferentially targets less differentiated memory T cells. The study should cite early work by John Harty's group showing that inflammatory signals reduce the establishment of CD8 T cell memory [PMID: 17579021].

As requested by this reviewer, we have revised the main text in the fourth paragraph of the Discussion (p. 14, highlighted in green) to cite and discuss an early paper by John Harty's group. However, we are concerned that this reviewer may have mentioned an irrelevant paper, PMID:17579021, since the role of IFN-g in the differentiation of memory CD8 T cells was not addressed in that paper. We believe that this reviewer was referring to another paper, PMID: 15247915, which showed that IFN-g deficiency favors the differentiation of antigen-primed CD8 T cells into CD127^{high} long-lived MP cells rather than short-lived TE cells. As this paper is more relevant to our current manuscript, we have cited and discussed this paper.

Secondly, analysis of the source of the IFN γ (colonic CD8 T cells predominantly) enables some speculation as to how loss of MHCII might drive this. The authors include some limited discussion of this, however there should be some discussion around regulatory T cells. Instinctively, one imagines that production of the proinflammatory cytokine IFN γ in MHCII^{-/-} mice might occur due to a lack of regulation. There are no Tregs in MHCII^{-/-} mice. Are there any in CD4^{-/-} mice?

We thank this reviewer for raising an important point. As suggested by this reviewer, we have now discussed the potential role of Treg cells in the regulation of IFN-g production by colonic CD8 T cells in the fifth paragraph of the Discussion (p. 14, highlighted in green). Indeed,

some Foxp3+ T cells with suppressive function develop in CD4-deficient mice, suggesting a role for these cells. Although some Foxp3+ T cells also develop in MHCII-deficient mice, they may be quantitatively or qualitatively different from those in CD4^{-/-} mice and may be unable to limit IFN- γ production in colonic CD8 T cells.

This is an excellent paper with important results. My only other suggestion would be to alter the title to reflect the more significant finding that persistence of CD8 T cell memory is not dependent on CD4 T cells, but rather on low levels of IFN γ . Or that loss of CD8 T cells in MHCII^{-/-} mice is not due to dependence on CD4 T cells, but to high IFN γ levels. As it stands the title states something that we've known for some time (i.e. that IFN γ can erode CD8 T cell memory).

We have followed this reviewer's advice and changed the title to "Memory CD8 T cells are vulnerable to chronic interferon- γ signals but not to CD4 T cell deficiency in MHCII-deficient mice".